# HYPERmotion: Learning Hybrid Behavior Planning for Autonomous Loco-manipulation

**Jin Wang[1][2][†], Rui Dai[1][2][†], Weijie Wang[1][2][†], Luca Rossini[1], Francesco Ruscelli[1], Nikos Tsagarakis[1]**

[1]Istituto Italiano di Tecnologia    [2]Universita di Genova    [†] Equal Contribution

**Abstract:** Enabling robots to autonomously perform hybrid motions in diverse environments can be beneficial for long-horizon tasks such as material handling, household chores, and work assistance. This requires extensive exploitation of intrinsic motion capabilities, extraction of affordances from rich environmental information, and planning of physical interaction behaviors. Despite recent progress has demonstrated impressive humanoid whole-body control abilities, they struggle to achieve versatility and adaptability for new tasks. In this work, we propose HYPERmotion, a framework that learns, selects and plans behaviors based on tasks in different scenarios. We combine reinforcement learning with whole-body optimization to generate motion for 38 actuated joints and create a motion library to store the learned skills. We apply the planning and reasoning features of the large language models (LLMs) to complex loco-manipulation tasks, constructing a hierarchical task graph that comprises a series of primitive behaviors to bridge lower-level execution with higher-level planning. By leveraging the interaction of distilled spatial geometry and 2D observation with a visual language model (VLM) to ground knowledge into a robotic morphology selector to choose appropriate actions in single- or dual-arm, legged or wheeled locomotion. Experiments in simulation and real-world show that learned motions can efficiently adapt to new tasks, demonstrating high autonomy from free-text commands in unstructured scenes. Videos and website: hy-motion.github.io/

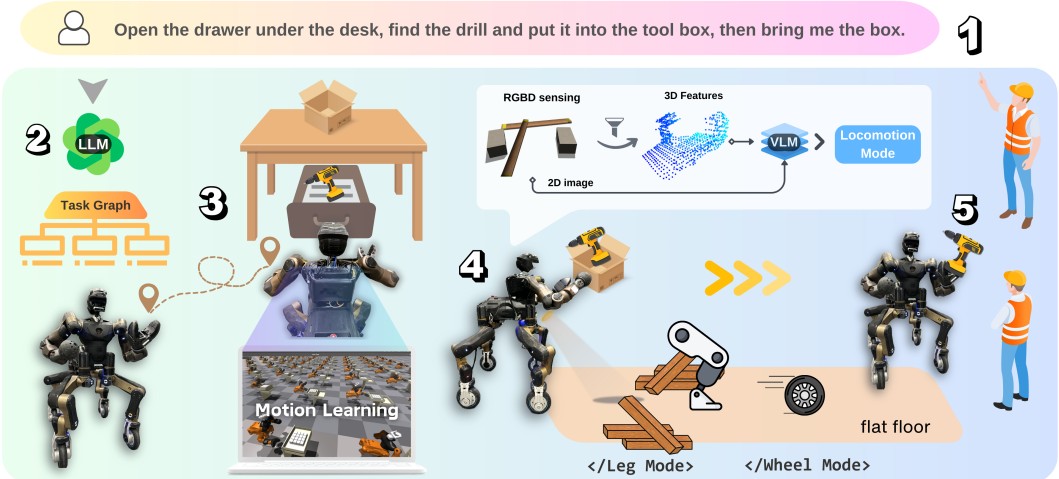

Figure 1: **HYPERmotion** enables the humanoid robot to **learn**, **plan**, and **select** behaviors to complete long-horizon tasks. Steps 1-5 illustrate how the robot, guided by foundation models, autonomously performs locomotion and manipulation after interpreting verbal instruction and chooses motion modes for different scenarios independently.

8th Conference on Robot Learning (CoRL 2024), Munich, Germany.

# 1 Introduction

Humanoid robots with behavioral autonomy have consistently been regarded as ideal collaborators in our daily lives and promising representations of embodied intelligence. Compared to fixed-base robotic arms, humanoid robots, due to their configuration characteristics, offer a larger operational space while significantly increasing the difficulty of control and planning. Despite the rapid progress towards general-purpose humanoid robots [1, 2], most studies remain focused on locomotion with few investigations into learning whole-body coordination, which results in simplistic control designs that struggle to adapt to new tasks and environments, thus limiting the potential to demonstrate long-horizon tasks under open-ended instructions. One major challenge is how to explore actionability of humanoid robots and diversify their behaviors, while learning to infer affordances and spatial geometric constraints, thus making plans for various tasks with learned skills like humans.

Recently, vibrant advances in robotics learning have made it a promising avenue for manipulation and locomotion [3, 4, 5, 6]. Learning-based methods, such as reinforcement learning (RL), have become effective tools for task-oriented action generation [7, 8, 9] while facilitating generalization across diverse scenarios. However, extending learning algorithms to humanoid robots remains a challenge stemming from the exponential increase in training costs induced by high degrees of freedom (DoF) and the difficulty of deployment on real robots under dynamic constraints. Meanwhile, the rise of large language models (LLMs) and their remarkable capabilities in robotic planning [10, 11, 12] have made it possible to perform logical reasoning and construct hierarchical action sequences for complex tasks. By integrating observations from different modalities, those models can be used for extracting features of objects and environments for robot perception and decision-making. Nevertheless, limitations to the utilization of LLM in humanoid robots exist, particularly in complex whole-body motion control and precise coordination between body parts.

To address these issues, we first recognized that directly outputting whole-body trajectories for a real-world multi-joint system through simulation training is inefficient and impractical. Therefore, we adopt a decomposed training strategy that modularly selects the actuation components related to given tasks, and project lower-dimensional space trajectory on the whole-body space with a unified motion generator. The trained actions are stored as skill units in the motion library. We utilize the LLM's ability to decompose complex semantic instructions consisting of multiple sub-tasks and design a modular user interface as model's input. The LLM selects skills from the motion library and arranges a sequence of actions, referred to as task graphs. Furthermore, the 3D features extracted from captured 2D images and depth data can be integrated with the visual language model (VLM) and robotic intrinsic characteristics, acting as a robotic motion morphology selector.

We refer to this study as **HYPERmotion**, a framework that tackles behavior planning for humanoid robot autonomous loco-manipulation using language models. By leveraging the interaction of distilled spatial geometry and 2D observation with VLM, it grounds knowledge to guide morphology selection combining robotic affordance. And bridge the gap between semantic space, robotic perception and action. We demonstrate a learning-based whole-body control to generate humanoid motion that adapts to new tasks and performs long-horizon tasks with primitive skills. We further illustrate through experiments how HYPERmotion can be learned and deployed on a high-DoF, hybrid wheeled-leg robot, performing zero-shot online planning under human instructions.

# 2 Related Work

**Planning and Reasoning via Language Model** Grounding pre-trained language models has become a promising avenue for robotic studies. Extensive prior works [13, 14, 15, 16] focus on planning and reasoning of robotic tasks with LLM, using it as a tool for code generation [10, 17], reward design [18, 19, 20], and interactive robotic learning [21, 22, 11, 23]. Several transformer-based architectural planners [24, 25, 26] showcase the potential for embodied usage of generating robot action. And with the integration of multi modalities such as visual and auditory [27, 28, 29, 30, 31], perception and behavior can be directly bridged with semantic commands. Further research involves creating a customized skill library[32, 33, 12] to link robot execution and high-level planning. A related line of works [34, 35, 36, 37] has also explored grounding affordances with foundation mod-

els to enable spatial reasoning and guide manipulation. However, most efforts focus on employment of fixed robotic arms, with few attempts made to extend language modes to humanoid robots, due to their complex dynamics and precise coordination between different components. [2] presents an end-to-end humanoid manipulation towards speech reasoning but lacks the cooperation of mobility. [38][39] use LLMs for decision making and learning, while demonstrating solely in simulation scenes. In contrast, we realize language model based online planning and humanoid motion bootstrapping, distilling spatial knowledge for robotic morphology selection using only onboard sensing.

**Task-orient Humanoid Control** As the practical value of general-purpose humanoid robots becomes evident, a substantial amount of research has focused on the hardware of humanoid robots [40, 41, 42], as well as gait generation and balance control [43, 44, 45, 46]. Methods based on learning and model predictive control (MPC) have significantly enhanced the mobility of such robots [47, 48, 49, 50, 51, 52, 53]. Some demonstrate motion generation through teleportation [54] and imitation learning [55], while these often lack autonomy and struggle to organize learned short-horizon skills. Recent works on long-horizon tasks [56, 57] and bimanual coordination [58, 59] show dexterity and stabilization, but these are often limited to specific tasks to utilize their characteristics. Our approach enables the robot to perform composite tasks involving locomotion and manipulation in unstructured environments, engaging in rich physical interactions with various objects. Our work can also decompose tasks into sub-modules based on verbal instructions and autonomously perform tasks using pre-trained motions.

**Learning-based Whole-body Motion** Recent research has demonstrated significant advancements in robust walking [60, 61], trotting [62, 63, 64], and parkour [7, 8, 9] for legged robots using end-to-end RL. The combination of learning-based locomotion policy with model-based manipulation of attached arm shows feasible whole-body motion on rough terrain [5]. However, most learning-based controllers are implemented on quadruped robots with few DoFs, while highly redundant humanoids are rarely addressed. For the latter, optimization-based control is still necessary to ensure the safety and adherence to constraints due to limited reactive frequencies. Nowadays, learning-based MPC demonstrates capabilities in system dynamics identification [65, 66], closed loop performance [67, 68, 69] and safety assurance [70, 71, 72]. [73] shows a whole-body MPC on legged manipulators, but the tasks are limited by manually defined trajectories. In this work, we leverage RL to enable a wide range of motion skills without relying on predefined trajectories and employ a low-level optimization-based controller to ensure the feasibility of whole-body motion.

## 3 Methodology

In this section, we illustrate how the **HYPERmotion** framework enables the humanoid robot to autonomously perform loco-manipulation guided by semantic instructions (Sec. 3.1). We then provide the method for task-orient whole-body motion learning policy and how we build the humanoid motion skills library (Sec. 3.2). We further describe how to achieve robotic morphology selection based on spatial reasoning by integrating multi-modality language models, and map the long-horizon task to hierarchical behavior structure using learned motions (Sec. 3.3).

### 3.1 Autonomous Loco-manipulation via HYPERmotion

To address the autonomous loco-manipulation challenge for complex robotic platforms such as humanoid robots. This work proposes a method to perform language-guided behavior planning, motion generation and selection towards different scenarios. As shown in Fig 2, we divide the pipeline into four interrelated sectors that are learned and deployed sim-to-real manner. The motion generation sector selects RL training configurations for specific tasks and conducts training in parallel. The trajectory obtained from the training is provided as a reference to the optimizer, which ultimately generates whole-body motion skills and the skills will be stored in the motion library. The user input sector contains a user interface as well as pre-defined basic prompts, function options, and motion library, all of which together constitute the textual material fed to the LLM. After receiving a command, the task planning sector first generates a hierarchical task graph that includes task logic, condition determinations, and actions using the LLM. Once the task graph is loaded, it is interpreted as a Behavior Tree to guide the robot and to pass actions to lower-level execution. When a task

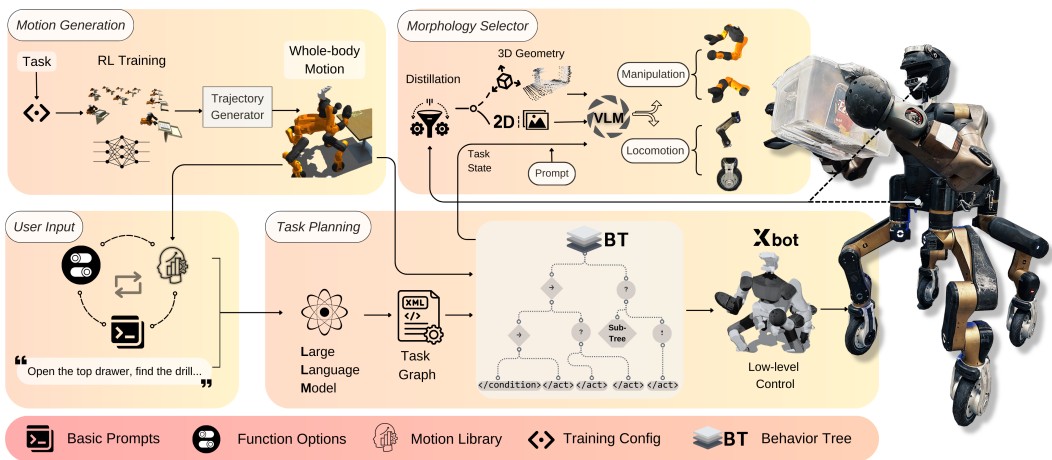

Figure 2: **Overview of HYPERmotion.** We decompose the framework into four sectors: **Motion generation** is assigned for learning and training whole-body motion skills for new tasks and storing them in the motion library. **User input** includes received task instructions and initialization prompt sets. **Task planning** generates a task graph that guides the robot's behavior through reasoning and planning features of LLM and passes action commands to the real robot. **Morphology Selector** is used for action determination in specific sub-tasks, selecting the appropriate morphology for locomotion and manipulation based on grounded spatial knowledge and robot intrinsic features.

requires selecting the robot's morphology, depth-sensing information is invoked and distilled into 2D images and geometric features. These data, along with the task state and prompts, are fed to the VLM, which then selects the morphology capable of achieving the goal. Through the coordination of these sectors, HYPERmotion facilitates semantic command understanding and zero-shot behavioral planning and action execution for humanoid robots, as shown in Fig 1.

## 3.2 Learning Whole-body Motion Generation

**Tasks Learning with RL** In this section, we show the details of learning whole-body motion generation based on different tasks, as shown in Fig. 3. To reduce the action space, we separate the robot's upper body from its legs. To guarantee feasibility, the floating base motion is heuristically limited to avoid generating unfeasible motions for the legs once projecting the trajectory on the whole-body space of the robot. For single-arm tasks, the action space $\mathcal{A}_1 \subseteq \mathbb{R}^{14}$ consists of the 6-DoF right arm joint angles, 6-DoF floating base translation distances and Euler angles, one torso yaw, and one gripper joint angle. The left arm is fixed during these tasks. For the dual-arm picking task, the action space $\mathcal{A}_2 \subseteq \mathbb{R}^{19}$ appends the 6-DOF left joint angles, with the gripper joint closed in this case. The observations include the states of the corresponding targets and the joint states of the robot's upper body, detailed in the Appendix along with the policy settings. All tasks utilize proximal policy optimization (PPO) [2] because of its efficiency. The output in the RL layer is a joint position trajectory $\mathbf{q}^* \in \mathbb{R}^{20}$ of the upper body. We train all skill policies separately using a general reward formulation:

$$r = \alpha_1 r_{l_{reach}} + \alpha_2 r_{r_{reach}} + \alpha_3 r_{rot} + \alpha_4 r_{finger} + \alpha_5 r_{task} + \alpha_6 r_{penalty} \tag{1}$$

where $r_{l_{reach}} = (\frac{1}{1+d_l^2})^2$ and $r_{r_{reach}} = (\frac{1}{1+d_r^2})^2$ with $d_l$ and $d_r$ representing the distance of the operational target to the left and right end effectors, respectively. The term $r_{rot} = \text{sign}(d_x) * d_x^2 + \text{sign}(d_z) * d_z^2$ is the reward for aligning the gripper's orientation with the task's object (e.g. drawer handle, door handle, drill). Here $d_x$ and $d_z$ are the dot products of the gripper's forward and up axes with the object's inward and up axes, respectively. The term $r_{finger} = \beta - (d_t + d_b)$ encourages the gripper to grasp the objects, where $\beta$ a fine-tuning parameter related to the size of the operational object, and $d_t$ and $d_b$ are the distances from the top and bottom links of the gripper to the task's object, respectively. $r_{penalty} = -\|\mathbf{a}\|^2$ penalizes excessive actions $\mathbf{a}$ to ensure smooth operation. Finally, $r_{task}$ denotes the specific reward for task completion, which will be detailed in the Appendix along with the specific settings of the parameters $\alpha_1$ to $\alpha_6$ and the axes for different tasks.

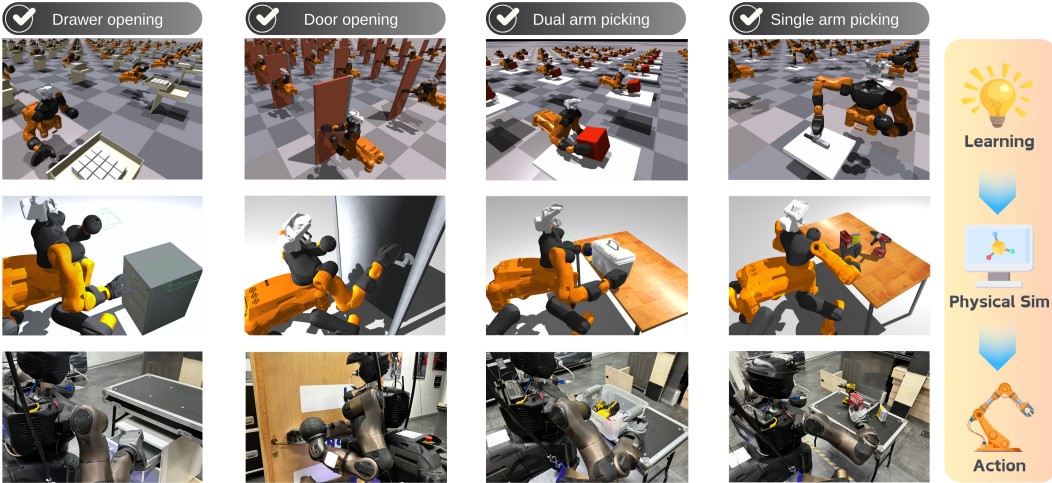

Figure 3: Whole-body tasks learning illustration in training, simulation and real-world settings.

**Whole-body Motion Generation** The reduced robot space from RL should be mapped into the whole body joint space to generated feasible trajectory. To fit the gap between the upper body trajectory and the whole body action, we solve an optimal control problem to merge them enforcing the whole-body dynamics of the robot to guarantee the feasibility of the resulting motion. In this case, we choose a unified whole-body trajectory generator[75], specifically, we use the framework presented in [75] to solve the following non-linear problem:

$$
\begin{cases}
\min_{\mathbf{x}(.),\mathbf{u}(.)} \int_0^T L(\mathbf{x}(t), \mathbf{u}(t), t)dt \\
\text{s.t. } \dot{\mathbf{x}}(t) = \boldsymbol{f}(\mathbf{x}(t), \mathbf{u}(t), t) \\
\boldsymbol{g}_1(\mathbf{x}(t), \mathbf{u}(t), t) = 0 \\
\boldsymbol{g}_2(\mathbf{x}(t), \mathbf{u}(t), t) \leq 0
\end{cases}
\tag{2}
$$

where $\mathbf{x}(t) = [\mathbf{q}, \mathbf{v}] \in \mathbb{R}^{n_x}$, $n_x = 93$ is the state number, $\mathbf{u}(t) = [\dot{\mathbf{v}}, \mathbf{f}_c] \in \mathbb{R}^{n_u}$, $n_u = 58$ are vectors of state and input variables, $\boldsymbol{q}, \boldsymbol{v}$ are the generalized coordinates and generalized velocities, $\mathbf{f}_c$ is foot force. $L(\mathbf{x}, \mathbf{u}, t) = \|\mathbf{q}^u - \mathbf{q}^*\|^2 + \|\mathbf{u}\|^2$ is intermediate cost, $\mathbf{q}^u$ is the upper body joint variables. As for constraints, $\dot{\mathbf{x}}(t) = \boldsymbol{f}(\mathbf{x}(t), \mathbf{u}(t), t)$ is the whole body dynamics, $\boldsymbol{g}_1$ represents the set of equality constraints, $\boldsymbol{g}_2$ the set of inequality constraints consisting of joint limits, velocity limits, and unilaterality of the contact forces. Finally, we could get the whole body motion combining upper body reference trajectory and whole body dynamic feasibility to complete the given tasks.

**Motion Library** After learning task-orient motion skills, we constructed a motion library to host these primitives, which consists of attribute and functional descriptions of these actions, and the corresponding learning-based whole-body policies. Then, the LLM can reason the attribute-function descriptions to create sequences of actions to be executed based on different tasks and generate a task graph to invoke the execution of each node without additional training or demonstration.

### 3.3 Humanoid Robot Task Planning with grounded language models

Migrating foundation models from a fixed robotic arm to a humanoid robot with a floating base presents numerous issues and challenges. The addition of robotic components not only imposes complex dynamic constraints, making it difficult to coordinate and control various parts. It also requires addressing the potential for different manipulation modes inherent in human-like structures, as well as the increased DoF for spatial mobility by the addition of wheels and legs. Due to the construction of our motion library, the usage of the LLM for planning no longer requires additional considerations for constraints such as self-collision or self-posture balance maintenance. This allows more focus on the decomposition of given tasks, and the selection of the robot's morphology.

**Humanoid Motion Morphology Selection** Humans utilize common sense and learned experiences to extract the affordances of objects they manipulate and select appropriate movement based on the estimation of geometric constraints of the environment. Inspired by this, we leverage VLM to

implement similar functionalities in robots. First, we include descriptions of the robot's structure and functions, and the robot's achievable range of motion in the prompts $\mathbf{p}_V$. While determining the morphology for a manipulation task $T_m$, the robot utilizes 2D and depth images from its head camera. Object detection and pose estimation algorithms [76, 77] are invoked to acquire the position and orientation of the target object $\mathbf{v}_c \in \mathbb{R}^6$, which is then transformed into the robot's coordinate system $\mathbf{v}_R \in \mathbb{R}^6$. The VLM $\mathcal{V}$, based on the current task state $\mathbf{s}$, the scene's 2D images $\mathbf{I}_{\text{scene}}^h$, and the target object's 6D pose $\mathbf{v}_R$, generates the robot's manipulation morphology $\mathbf{x}_m$ for the task scenario. For locomotion tasks, the robot uses the depth information from its pelvis depth camera to generate the point cloud $\mathbf{P}_c$, which is down-sampled to create a voxel grid $\mathbf{V}_g$. This spatial information containing the current moving path together with the 2D image $\mathbf{I}_{\text{scene}}^p$ and the task state are finally fused to select the robot's locomotion morphology $\mathbf{x}_l$ using the VLM $\mathcal{V}$.

$$\mathbf{x}_m = \mathcal{V}(\mathbf{s}, \mathbf{I}_{\text{scene}}^h, \mathbf{v}_R, \mathbf{p}_V) \tag{3}$$

$$\mathbf{x}_l = \mathcal{V}(\mathbf{s}, \mathbf{I}_{\text{scene}^p}^p, \mathbf{V}_g, \mathbf{p}_V) \tag{4}$$

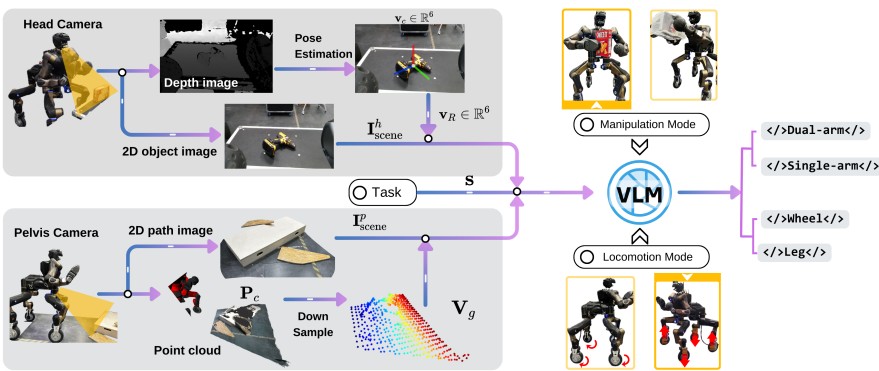

Figure 4: **Robotic Morphology Selector** extracts spatial geometric data and 2D observations from the physical environment upon receiving the language-conditioned task state and interacts with the VLM incorporating the grounded robot's affordances, so as to provide the optimal motion morphology that meets the requirements of given task scenario during manipulation and locomotion process.

**Zero-Shot Robot Behavior Planning** To obtain the desired response from LLM, it is necessary to impose constraints on the input. In the user interface, we define three types of constraints. The basic prompt provides a description of task background and characteristics of the robot, as well as interpretation of the user command and the output format. The motion library offers a catalog of learned skills and their description. Function option module offers specifications of the added functions developed for humanoid robot and determines whether these predefined functions are invoked during planning. Such as, if the morphology selection is chosen, the LLM will incorporate the morphology selector based on the task scenario; otherwise, this function will not be considered (See the Appendix). This approach allows for systematic construction of prompts and modular addition of constraints, thereby enhancing the flexibility of planning. We utilize BehaviorTree (BT) [78] as an intermediate bridge to convert high-level instructions into executable low-level skill sequences. BT provides a hierarchical structure for guiding actions and making decisions for the robot, which is composed of nodes with different effects. With a pre-defined motion library, LLM can generate a task graph consisting of learned motions and BT nodes, build it in an XML file, which constructs the complete BT. Thus realizing the robot's behavior planning with LLM by giving verbal instruction.

## 4 Experiment

We demonstrate HYPERmotion's ability to learn, plan, and select behaviors for different tasks in both simulation and real-world experiments using objects that can be commonly found in daily life. Our robot is a centaur-like humanoid robot, supported by four legs with wheels. The robot has two arms with one claw gripper on its right arm. There are two depth cameras, one is on the head and another is in the pelvis position. We use Xbot [79] to achieve real-time communication between the underlying actuators and the control commands. We use Isaac Gym [80] as a training environment

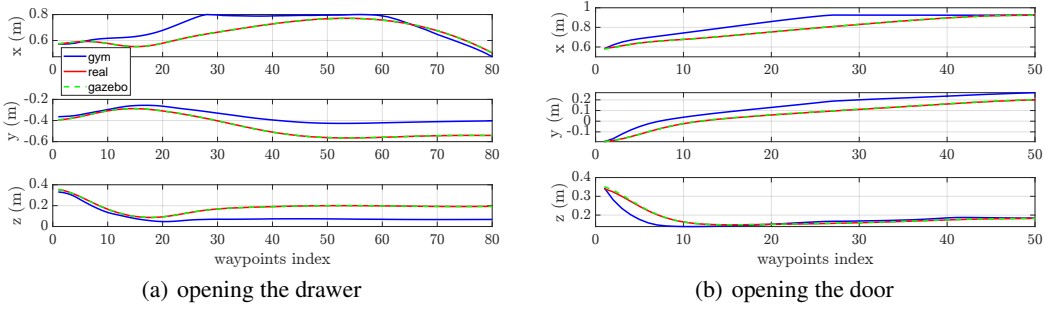

(a) opening the drawer       (b) opening the door

Figure 5: End effector position trajectory when executing different tasks in various environments

and validate the planned whole-body motion using Gazebo [81]. We access the `gpt-4o` model as the LLM planner and the `gpt-4v` model as the VLM from OpenAI API [82].

We trained individual motion primitives based on everyday tasks, deploying these skills on the robot to verify their feasibility (Sec. 4.1). We tested the adaptability and versatility of morphology selector in various task scenarios (Sec. 4.2) and conducted further study to validate the behavioral planning capability and performance of our framework in response to open-ended instructions (Sec. 4.3).

## 4.1 Whole-body Trajectory Learning

We pick two representative motions and compare the trajectories in Isaac Gym, Gazebo and the real world, respectively, as illustrated in Fig. 5, using the end effector position trajectories as an example. In the drawer opening task, the end effector first approaches the drawer handle and then pulls it out, demonstrating an initial increase followed by a decrease in the x-axis distance. In the door opening task, the end effector reaches the door handle and pushes it down. Self-collision is not considered during the training period because we still need to account for collisions between the upper body and legs, causing the arm to reach the target point at a faster speed. Instead, the whole-body planner projects the trajectories within constraints to avoid self-collision of the entire body. The trajectories from training are effectively tracked, with smoother results achieved after applying the whole-body controller as a filter. This indicates the successful deployment of our methods on the real robot.

## 4.2 Morphology Selection Towards Different Scenarios

We investigated whether a VLM can zero-shot determine robot's morphology based on task scenarios. We picked ten scenarios each for manipulation and locomotion in both simulation and real-world environments (See the Appendix). We compared the success rate of the VLM's morphology selection using 2D image input only versus image combined with spatial geometric as input. Each scenario was tested 10 times under both inputs, as shown in Fig. 6. We found the morphology selector effectively chooses the optimal mode for everyday object manipulation and mobile environment with a high average success rate. Compared to solely image input, adding spatial information improves the selector's accuracy, particularly in determining locomotion modes and adapting to complex scenarios (paths with obstacles of varying types and heights), thus leveraging the robot's affordances and leading to robust execution.

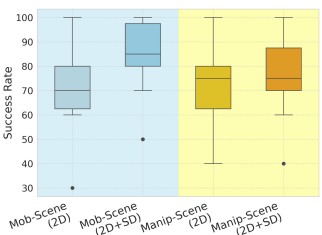

Figure 6: Success rate of the morphology selector for different scenarios. "2D" and "SD" are image and Spatial Data inputs.

## 4.3 Loco-manipulation Tasks with Language Model Planner

**Tasks with human instructions** We validate the ability of LLM to plan motion primitives for different loco-manipulation tasks, as well as the effect of the modular user input designed for humanoid robots regarding reasoning and planning. Experiments were conducted on tasks requiring a combination of perceptions and actions. We recorded the success rate and the impact of different errors of 4 representative tasks and provided quantitative evaluations in Fig.8. The results show the LLM based planner can effectively plan semantic instructions based on learned skills and guide the robot to complete a variety of tasks according to the action sequences, achieving a desired success rate

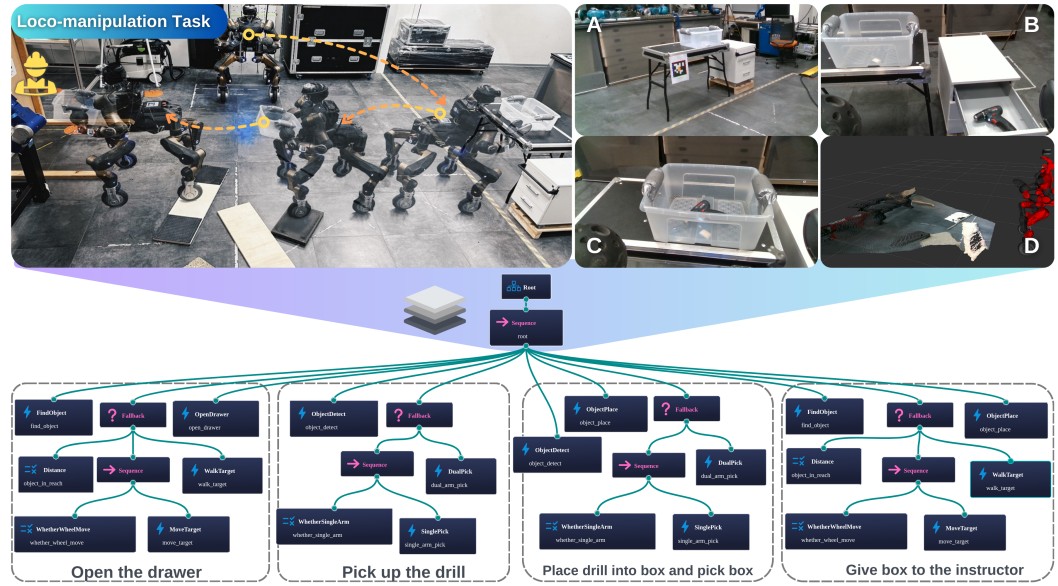

Figure 7: **Overall look of the long-horizon task** Images above show the timelapse of roll outs to robot motion trajectory. A. semantic navigation using AprilTag. B. object detection and pose estimation. C. manipulation morphology selection. D. locomotion morphology selection. The BehaviorTree below shows the details of LLM task planning.

($\geq 60\%$) on real-world robots. Whereas adding task complexity and selecting multiple functional modules as input increases the difficulty of planning, and execution errors mainly stem from intricate dynamical constraints on the actions and misalignment of the floating sensing with robot execution.

**Long-horizon Task** We further explored whether HY-PERmotion can enable behavior planning for a humanoid robot towards long-horizon tasks. We orchestrated a collaborating task scenario and input verbal instruction as shown in Fig.1. Qualitative results including time-lapse shots of robot motion execution and a Behavior Tree mapped out by LLM are shown in Fig.7. We demonstrate that our framework can synthesize sequences of motion primitives based on designed user input and accurately infer the logic of semantic knowledge while selecting robotic morphology of locomotion and manipulation according to the environment and state of the task. We found that language-based behavior planner exhibits greater versatility and adaptability to more complex tasks compared to existing methods.

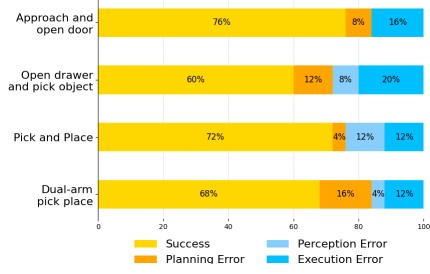

Figure 8: Average rate of the humanoid robot successfully performing various LLM planning tasks, and failure caused by different type of errors during the tasks.

## 5 Conclusion

We present **HYPERmotion**, a framework that enables humanoid robots to learn, select, and plan behaviors, integrating knowledge and robotic affordance to perform embodied tasks. We evaluate the framework's efficiency and versatility through real-world experiments and long-horizon tasks. Despite achieving expected results, there are **limitations**: the motion library's size restricts the range of task commands, and learning of new skills requires separate training optimization, hindering generalization from existing actions. Moreover, the system struggles to handle external disturbances and collisions, lacks real-time linguistic interaction during the task and has limited capability for re-planning in response to unexpected tasks. Future work will focus on enriching the robot's action skills, enhancing LLM dynamic planning ability, and improving robot navigation and perception to achieve close-loop humanoid motions and safe human-robot collaboration.

**Acknowledgments**

This work was supported by the European Union's Horizon 2020 research and innovation programme, euROBIN EPUE034001.

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

# Appendix

# Contents

## A1  Robot System Setup

### A1.1  Robot hardware

Our robot is a centaur-like robot platform. The upper body of the robot is humanoid in design and is similar in size to the average human to adapt to both dual-arm and single-arm manipulation. The robot's mobility relies on its quadrupedal lower body and maintains whole-body balance to cope with a variety of terrain conditions and perform loco-manipulation tasks. Moreover, to improve the robot's mobility on flat ground, wheel modules are integrated underneath each leg and can control the direction and steering of the wheels.

The robot's whole body consists of 38 actuatable joints. The robot's torso is mounted on the pelvis of the lower body via yaw joints, allowing the upper body to rotate in the transverse plane. Each arm of the robot includes 6 DoF, where the right hand gripper contains one extra DoF that controls its opening and closing. The robot's legs are designed to provide an omni-directional wheeled motion and articulated legged locomotion, with each leg containing six degrees of freedom, allowing for positioning, orientation, and rotation of the wheeled-leg module.

The perception system of the robot consists of two on-board RealSense Depth Camera D435i, one located in the robot's head and the other in the robot's pelvis, which are used to provide 2D images and depth information of the surrounding environment and objects. The complete computing system consists of two on-board computing units (ZOTAC-EN1070K PC, COM Express conga-TS170) for system communication and real-time robot control and an external pilot PC (Inter Core i9-13900HX CPU @3.90GHz, NVIDIA GeForce RTX 4090) for task planning and sensory data processing as well as a user interface.

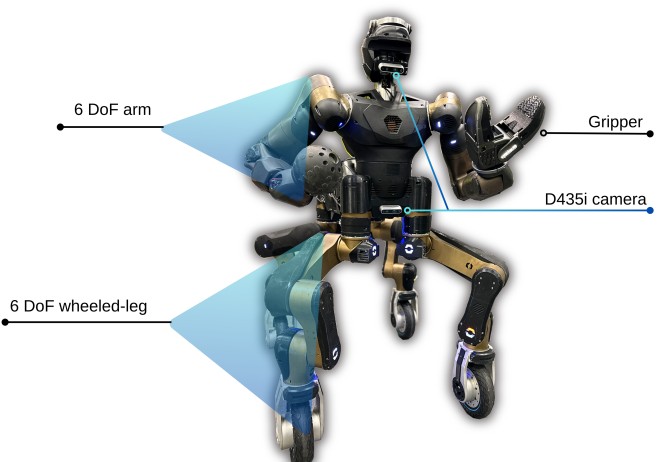

Figure A1: Robot hardware setup

### A1.2  Robot software

We use XBotCore, a cross-platform, real-time, open-source software designed for interfacing with low-level hardware components of robots [1]. This innovative tool enables effortless programming and management of various robotic systems by offering a standardized interface that conceals the intricacies of the hardware. Additionally, a proprietary CartesI/O motion controller [2] handles higher-order motion instructions. It is capable of managing multiple responsibilities and restrictions, prioritized according to the demands of specific situations. Through solving a series of quadratic programming (QP) challenges, each linked to a unique priority tier, the controller ensures optimal performance across all preceding priority stages.

## A2  Additional Experiment and Evaluation

### A2.1  Experiment Setup

We experimentally validate the efficacy of HYPERmotion in behavior learning, planning, and decision-making by implementing the framework and evaluating its performance on a wheel-legged humanoid robot executing long-horizon tasks in response to semantic commands.

We conducted the experiment using objects that are commonly found in an office kitchen. The test environment was an open area inside the lab, with objects randomly placed. The AprilTag system [3], which incorporates a vision-driven algorithm, was used during the long-horizon task to identify the relative objects' location and direction of recognized tags. Within the actual environment, we employ AprilTags to gather task-specific observations. A single visual marker on the door allows for the determination of the door handle's relative position. The robot searches for the tag if it doesn't exit the camera's field of view (FOV). Additionally, AprilTags enable the identification of the drawer's relative positions.

### A2.2  LLM based Task Planner

**Long-horizon Task**

A long-horizon task refers to a task that needs to be completed over an extended time period, typically involving the execution of multiple subtasks and decision-making across several steps. Such tasks usually require the system to possess strong planning capabilities and long-term strategies to effectively achieve objectives in complex environments. In this study, we define long-horizon tasks based on the number of robotic actions included in the task. When the task described by the given instructions comprises four or more robotic actions, we classify it as a long-horizon task. Here, robotic actions refer to those included in the action library. For example, the task 'Pick the box' consists of three actions: $< ObjectDetect >$, $< MoveTarget >$, and $< DualPick >$, and is thus categorized as a simple motion task. In contrast, the long-horizon task described in Fig. 1 requires the robot to perform multiple actions, including $< FindObject >$, $< MoveTarget >$, $< ObjectDetect >$, $< SinglePick >$, and $< OpenDrawer >$ et., with a longer time span required for task completion.

**Behavior Planner**

We initially conducted simulation experiments to compare the construction of behavior trees using traditional method [4] with LLM based planner. The experiments included a qualitative analysis of the capabilities of both approaches, as well as the planning results for various robotic tasks. Executable rate refers to whether the generated BT is logically consistent and executable by the robot. Compared to normal BT-planner, the integration of LLMs' reasoning and decision-making capabilities enables the direct generation of BTs from tasks described in free-text instructions. Moreover, BT can be restructured by simply altering the input instructions, significantly enhancing the autonomy of task planning. Quantitative analysis in Table A1 indicates that BTs generated by LLMs exhibit higher executability and task success rates compared to those generated by BT-planner. Additionally, the time required for robots to complete tasks in the simulation environment was reduced. These results underscore the advantage of LLMs' reasoning and text generation capabilities in constructing behavior trees, which rely on a deep understanding of task logic and adherence to fixed construction rules.

**Loco-manipulation Tasks Simulation Results**

After verified that the LLM-based planner is capable of effectively planning robotic tasks, we selected six loco-manipulation tasks to validate the performance and adaptability of the HYPERmotion framework performing different tasks. We compared its performance with traditional whole-body MPC control methods [5] and conducted 25 experiments for each tasks separately, the simulation results are shown in Table A2.

Table A1: Comparison of different methods for behavior planning

| Method | Abilities | | | Loco-manipulation Task | | | Long-horizon Task | | |
|---|---|---|---|---|---|---|---|---|---|
| | Autonomy | Replan | Text Input | Exec ↑ | Succ ↑ | Avg. Time ↓ | Exec ↑ | Succ ↑ | Avg. Time ↓ |
| BT-Planner | *low* | ✗ | ✗ | 92% | 80% | 39.07s ± 12.4 | 74% | 68% | 145.37s ± 32.6 |
| LLM-BT | *high* | ✔ | ✔ | 98% | 94% | 32.79s ± 8.6 | 86% | 82% | 121.20s ± 20.4 |

Table A2: Comparison of different methods towards various robotic tasks. TLE(time limit exceeded)

| Task | WB-MPC | | LLM-Planner | | LLM-Planner+MS | |
|---|---|---|---|---|---|---|
| | Succ ↑ | Avg. Time ↓ | Succ ↑ | Avg. Time ↓ | Succ ↑ | Avg. Time ↓ |
| Move to target | 84% | 25.7s ± 12.3 | 96% | 36.8s ± 10.6 | 92% | 40.1s ± 29.3 |
| Approach Object | 72% | 24.3s ± 10.6 | 88% | 34.9s ± 8.6 | 96% | 39.8s ± 22.1 |
| Open door | 64% | 45.6s ± 13.5 | 84% | 35.2s ± 12.6 | 88% | 42.5s ± 18.6 |
| Pick object | 68% | 38.9s ± 24.2 | 80% | 38.6s ± 9.6 | 84% | 44.3s ± 22.3 |
| Pick and place object | 60% | 50.2s ± 47.3 | 72% | 49.2s ± 23.6 | 84% | 54.7s ± 32.1 |
| Open drawer and pick object | 0% | TLE | 64% | 105.4s ± 27.6 | 72% | 126.3s ± 33.6 |

The experimental results indicate that using an LLM to plan pre-trained motion primitives for executing loco-manipulation tasks yields a higher success rate compared to WB-MPC, particularly for complex tasks such as 'pick and place object' and tasks requiring long-term planning such as 'open drawer and pick object.' However, for some simpler tasks, the time required by the LLM-Planner is slightly longer than that of WB-MPC, attributable to the time taken for online API requests and BT construction. Adding Morphology Selector (MS) into the task planning process results in a modest increase in the average planning time, but it concurrently enhances the success rate of tasks, especially those involving interaction with the surrounding environment and objects.

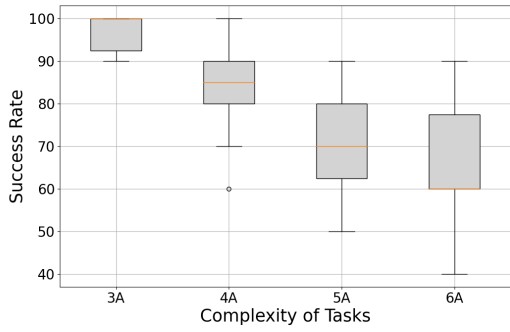

Figure A2: Success rate of loco-manipulation tasks in relation to task complexity.

We further investigated the relationship between the success rate of executing loco-manipulation tasks using HYPERmotion and the complexity of the tasks. We conducted ten separate simulation experiments for tasks involving 3, 4, 5, and 6 actions. The results show that as task complexity increases, the success rate of task completion gradually decreases, though the overall average success rate remains above 60%. Additionally, task success rates vary depending on the specific content of the tasks and the difficulty of the required robotic actions, with this variability being more pronounced in long-horizon tasks (those involving four or more actions).

## A2.3 Failure Cases Study

During the real-world experiment, failure of different cases happened in the process of the robotic tasks performing. In Fig. 8, we recorded the average rate of failure caused by different type of errors. Planning errors primarily occur during the BT construction process by the LLM-Planner. When tasks are highly complex and require multiple estimations of the environment or object states, the task graph generated by the LLM may exhibit logical errors or missing actions, leading to task failure. To address this issue, we optimized the motion primitives by integrating the robot's self-state detection or environmental perception into the primitives to reduce the logical complexity of constructing BT.

Perception errors stem from inaccuracies in the robot's sensing system, particularly in tasks involving navigation and target or object detection. The perception system consists of AprilTag localization and deep object pose estimation. While using an RGB camera for AprilTag localization, there is a margin of error of 2-4 cm, especially when the robot is in motion. The accuracy of the object pose detection algorithm is also influenced by the object's orientation and potential occlusions, which can affect the robot's manipulation posture.

When performing robotic tasks in the real world, execution errors are the most common and can arise from a variety of factors. During task execution, since the actions are pre-trained and optimized for full-body motion, unexpected disturbances can significantly impact performance. For instance, while opening a door, if the robot's leg collides with the doorframe, it may struggle to adapt its trajectory in real-time. In the case of long-horizon tasks, errors tend to accumulate with the increasing number of actions performed. For example, after pulling open a drawer and retrieving an object, any deviation in the grasping pose can lead to failure in placing the object, causing it to collide with the environment and drop. Additionally, tasks with extended durations can result in overheating of the robot's actuators, further compromising task performance.

Applying LLMs to a high-DOF, floating-based humanoid robot, where manipulation must be coordinated with locomotion, is a complex and challenging endeavor. Due to the low real-time performance of LLMs, they are unable to update plans in real-time during task execution and cannot directly generate control trajectories for multiple joints simultaneously. To address this, we consider the integration of a task graph and motion library, where the LLM is responsible for reasoning and decision-making, constructing a task graph composed of motion primitives. This task graph then issues control commands to enable the robot to complete long-horizon tasks.

During the acquisition of motion primitives, the prioritization in the whole-body motion optimization phase significantly influences action generation. For instance, in actions like 'opening door' and 'open drawers', the optimization must not only account for the accuracy of the motion trajectory but also maintain robot balance and avoid self-collision. Additionally, to accommodate drawers of varying heights, we incorporate floating base adjustments in the optimization process.

When dealing with long-horizon tasks, differences in pose solving can lead to issues with action continuity. For example, in the "Pick and place object" task, we introduced a homing action between the pick and place phases to ensure seamless transitions between actions to reduce the error rate.

## A3 Details of Robot Learning

We utilize Proximal Policy Optimization (PPO) [6] for training our tasks, employing a multi-layer perceptron within an actor-critic framework. The network architecture for the drawer opening, door opening, and dual-arm picking tasks consists of layers with [256, 128, 64] units while the picking task uses layers with [256, 128, 64] units. The activation function applied across all tasks is ELU. Below, we detail the observations, task-specific rewards ($r_{task}$), and reward parameters for each task in Sec. A3.1 and the task evaluation in Sec. A3.2

### A3.1 task design details

### A3.1.1 drawer opening

First, we define the frame of the drawer handle. The x-axis of the handle points towards the robot, while the z-axis points upwards. The handle's inward direction is aligned negatively along the x-axis, and the upward direction is consistent with the z-axis. The task reward is defined as

$$r_{task} = \alpha_7 r_{around} + l_{drawer} * r_{around} + l_{drawer} \tag{A1}$$

where $r_{around} = 0.5$ when the gripper's top link is above the handle's position and the bottom link is below the handle's position, otherwise $r_{around} = 0$. $l_{drawer}$ represents the length by which the drawer has been pulled.

The observations and reward parameters for this task are listed in Tab. A1 and A2.

| |
| --- |
| normalized upper body joints position |
| upper body joints velocity * 0.1 |
| drawer pulled length |
| vector from gripper to drawer handle |

| | |
| --- | --- |
| $\alpha_1$ | 2.0 |
| $\alpha_2$ | 0.0 |
| $\alpha_3$ | 0.5 |
| $\alpha_4$ | 7.5 |
| $\alpha_5$ | 7.5 |
| $\alpha_6$ | 0.01 |
| $\alpha_7$ | 0.7 |
| $\beta$ | 0.04 |

Table A3: observations of drawer opening task

Table A4: reward parameters of drawer opening task

### A3.1.2 door opening

The door handle has the same frame as the drawer handle. The task reward is defined as

$$r_{task} = \alpha_7 r_{around} + angle_{handle} * r_{around} + angle_{handle} + angle_{door} \tag{A2}$$

where $r_{around}$ is the same setting as the drawer opening task and $angle_{handle}$ represents the angle by which the door handle has been pushed. $angle_{door}$ is the angle of the opened door.

The observations and reward parameters for this task are listed in Tab. A3 and A4.

| |
| --- |
| base pose |
| right arm joints position |
| door handle pose |
| gripper pose |
| door handle angle |
| door opened angle |

| | |
| --- | --- |
| $\alpha_1$ | 2.0 |
| $\alpha_2$ | 0.0 |
| $\alpha_3$ | 1.5 |
| $\alpha_4$ | 7.5 |
| $\alpha_5$ | 2.0 |
| $\alpha_6$ | 0.01 |
| $\alpha_7$ | 0.125 |
| $\beta$ | 0.02 |

Table A5: observations

Table A6: parameters

### A3.1.3 single arm picking

We define the object's upward direction as aligning negatively along the x-axis, and the inward direction as aligning negatively along the z-axis. This orientation encourages the gripper to adopt a top-to-bottom pose, facilitating a proper grasp of the object. The task reward is defined as

$$r_{task} = \alpha_7 r_{around} + h \tag{A3}$$

where $r_{around}$ is the same setting as the previous tasks with the corresponding object frame and $h = 1$ if the object is been picked up, otherwise $h = 0$.

The observations and reward parameters for this task are listed in Tab. A5 and A6.

| | |
|---|---|
| $\alpha_1$ | 7.5 |
| $\alpha_2$ | 0.0 |
| $\alpha_3$ | 5.0 |
| $\alpha_4$ | 2.5 |
| $\alpha_5$ | 7.5 |
| $\alpha_6$ | 0.01 |
| $\alpha_7$ | 0.7 |
| $\beta$ | 0.1 |

| |
|---|
| base pose |
| right arm joints position |
| object pose |
| gripper pose |


Table A7: observations

Table A8: parameters


### A3.1.4 dual arm picking

In the dual arm picking task, the distance $d_l$ and $d_r$ represents the left end-effector and right end-effector to the left and right side of the object, respectively. The task reward is defined as

$$r_{task} = h \tag{A4}$$

where $h = 1$ if the object is been picked up, otherwise $h = 0$.

The observations and reward parameters for this task are listed in Tab. A7 and A8.

| |
|---|
| base pose |
| two arms joints position |
| object pose |
| left end-effector pose |
| right end-effector pose |
| vector from object left side to left end-effector |
| vector from object right side to right end-effector |

| | |
|---|---|
| $\alpha_1$ | 2.0 |
| $\alpha_2$ | 2.0 |
| $\alpha_3$ | 0.0 |
| $\alpha_4$ | 0.0 |
| $\alpha_5$ | 7.5 |
| $\alpha_6$ | 0.01 |
| $\alpha_7$ | 0.0 |
| $\beta$ | 0.0 |

Table A9: observations

Table A10: parameters

### A3.2 task evaluation

We evaluate the learning results by applying the trained policy with 2000 rollouts for each task across four different position randomization scopes. We train and evaluate under the same randomization scope. Except for single arm picking task, which randomizes the position of the object, other tasks randomize the position of the robot. We summarize the success rate of each task in Tab. A9.

| | mean relative position of robot's base to goal's origin (m) | domain of position randomization (m) | success rate |
|---|---|---|---|
| drawer opening | $[1.2, -0.4, 0.65]^\top$ | $[0.01, 0.10]$ | $77.8\% \pm 11.0\%$ |
| door opening | $[1.0, -0.1, 1.0]^\top$ | $[0.02, 1.0]$ | $85.3\% \pm 19.6\%$ |
| single arm picking | $[0.45, -0.3, 0.8]^\top$ | $[0.2, 0.5]$ | $88.5\% \pm 6.9\%$ |
| dual arm picking | $[0.9, 0.0, 0.8]^\top$ | $[0.02, 1.0]$ | $73.8\% \pm 15.7\%$ |

Table A11: success rate of each task under different position randomization scope

## A4 Details of Whole-body Optimization

The trajectory optimization problem essentially constitutes a Nonlinear Programming (NLP) challenge characterized by a predetermined quantity of nodes and intervals. Its canonical formulation

typically adheres to Eq.(A5)

$$\begin{cases} \min_{\mathbf{x}(.),\mathbf{u}(.)} \int_0^T L(\mathbf{x}(t),\mathbf{u}(t),t)dt \\ \text{s.t. } \dot{\mathbf{x}}(t) = \boldsymbol{f}(\mathbf{x}(t),\mathbf{u}(t),t) \\ \boldsymbol{g}_1(\mathbf{x}(t),\mathbf{u}(t),t) = 0 \\ \boldsymbol{g}_2(\mathbf{x}(t),\mathbf{u}(t),t) \leq 0 \end{cases} \tag{A5}$$

### A4.1 Formulation

The standard formulation(A5) necessitates conversion into a discrete programming format . Subsequently, we discrete the state and input variable as the follow sets, $N$ is the node number

$$\mathcal{X} = \begin{bmatrix} \mathbf{x}_1 \\ \vdots \\ \mathbf{x}_N \end{bmatrix} ; \mathcal{U} = \begin{bmatrix} \mathbf{u}_1 \\ \vdots \\ \mathbf{u}_N \end{bmatrix} \tag{A6}$$

then the general optimization form Eq.(A5) becomes Eq.(A7)

$$J = \sum_{i=0}^N L_i(\mathbf{x}_i,\mathbf{u}_i)$$
$$\dot{\mathbf{x}}_i = \boldsymbol{f}(\mathbf{x}_i,\mathbf{u}_i) , i = 0, \cdots N \tag{A7}$$
$$\mathbf{C}_{\min} \leq \mathbf{C}(\mathbf{x}_i,\mathbf{u}_i) \leq \mathbf{C}_{\max}, i = 0, \cdots N$$

where , $\mathbf{C}(\mathbf{x}_i,\mathbf{u}_i)$ is the discrete form of equality and inequality constrain, $\mathbf{C}_{\min}$ is the lower limit, $\mathbf{C}_{\max}$ is the upper limit.

### A4.2 Dynamic Equation

In order to keep the motion critical dynamic feasible, we construct our robotic's motion equation as following the whole body dynamic:

$$\dot{\mathbf{x}}_i = \begin{bmatrix} \dot{\mathbf{v}}_i \\ M(\mathbf{q}_i)^{-1} \left( \boldsymbol{J}_c^T(\mathbf{q}_i)\mathbf{f}_c^i - \boldsymbol{h}(\mathbf{q}_i,\mathbf{v}_i) + \boldsymbol{S}\boldsymbol{\tau}_i \right) \end{bmatrix}$$

where $\mathbf{x} = [\mathbf{q},\mathbf{v}] \in \mathbb{R}^{n_x}$ is the state number, $\mathbf{u} = [\dot{\mathbf{v}},\mathbf{f}_c] \in \mathbb{R}^{n_u}$ are vectors of state and input variables, $\mathbf{q},\mathbf{v}$ are the generalized coordinates and generalized velocities, $\mathbf{f}_c \in \mathbb{R}^{n_c}$ the vector of contact forces, $\boldsymbol{\tau}_i \in \mathbb{R}^{n_a}$ the vector of actuated joint torques, $M \in \mathbb{R}^{n_v \times n_v}$ is the inertia matrix, the $n_v$ is the dimension of general velocity, $\boldsymbol{h} \in \mathbb{R}^{n_v}$ the non-linear bias terms accounting for gravity, Coriolis and centrifugal torques, $\boldsymbol{J}_c \in \mathbb{R}^{n_c \times n_v}$ is the contacts Jacobian, matrix $\boldsymbol{S} \in \mathbb{R}^{n_v \times n_a}$ is used to map actuated torques to the full vector of efforts.

### A4.3 Feasible Constrain

Specifically, in order to keep the trajectory physical feasible, we should shape the constrains as the following:

1) Initial state: In order to safely execute the trajectory, the robotic should start acting from a initial state which has stability and safety. The initial state constrains are constructed as the flowing:

$$\mathbf{q}^0 = \mathbf{q}_{\text{init}} \text{ initial position} \tag{A8}$$

$$\mathbf{v}^0 = 0 \text{ initial velocity} \tag{A9}$$

2) State limitation: Before we have got the joints trajectory from RL, which has a rough feasible, so it is neccessary to consider the physical limit state of robot. The state limitation constrains are constructed as the flowing:

$$\mathbf{q}_{\min}^k \leq \mathbf{q}^k \leq \mathbf{q}_{\max}^k \text{ position bounds } \forall k \in [1, N-1] \tag{A10}$$

$$\mathbf{v}^k \leq \mathbf{v}^k \leq \mathbf{v}_{\max}^k \text{ velocity bounds } \quad \forall k \in [1, N-1] \tag{A11}$$

$$\dot{\mathbf{v}}_{\min}^k \leq \dot{\mathbf{v}}^k \leq \dot{\mathbf{v}}_{\max}^k \text{ acceleration upper bounds } \quad \forall k \in [0, N-1] \tag{A12}$$

$$\dot{\mathbf{v}}_{\min}^k \leq \dot{\mathbf{v}}^k \leq \dot{\mathbf{v}}_{\max}^k \text{ acceleration lower bounds } \quad \forall k \in [0, N-1] \tag{A13}$$

3) Contact Condition: As mentioned before, our robot has two strategies legged and wheel. So if just in the wheel motion, we should just keep the following constrain which keep the force reasonable:

$$\mathbf{f}_{c,j}^{z,k} \cdot \mathbf{n}_i > 0 \tag{A14}$$

if the robotic motion model is legged, we should keep the robotic not slip just as the constrain of friction cone (A15), so the contact condition should be (A14) and (A15) :

$$\left\| (\mathbf{f}_{c,j}^{x,k}, \mathbf{f}_{c,j}^{y,k}) \right\|_2 \leq \mu_i \left( \mathbf{f}_{c,j}^{z,k} \cdot \mathbf{n}_i \right) \text{ leg contact force bounds } \quad \forall k \in [0, N-1] \tag{A15}$$

where $\mathbf{f}_{c,j} = [\mathbf{f}_{c,j}^x, \mathbf{f}_{c,j}^y, \mathbf{f}_{c,j}^z]$, $\mathbf{j}$ is the $j$-th leg contact force, $j = 1, 2, 3, 4$.

4) Torque limitation: The underlying foundation of robot motion action is depend on the ability of joints motor, so it is important to keep the torque in the capability limitation of motor, just as the following (A16) and (A17):

$$\mathbf{S}\boldsymbol{\tau}_i = \mathbf{M}(\mathbf{q})_i \dot{\mathbf{v}}_i + \mathbf{h}(\mathbf{q}_i, \mathbf{v}_i) - \mathbf{J}_c^T(\mathbf{q}_i)\mathbf{f}_c^i \tag{A16}$$

$$\boldsymbol{\tau}_{fb}^k = \mathbf{0} \quad \forall k \in [0, N] \; \boldsymbol{\tau}_{j\,\min}^k \leq \boldsymbol{\tau}_j^k \leq \boldsymbol{\tau}_{j\,\max}^k \quad \forall k \in [0, N] \tag{A17}$$

where $\boldsymbol{\tau}_{fb}^k$ is the virtual float joints keep zero, $\boldsymbol{\tau}_j^k$ is the actuated joints torque, $\boldsymbol{\tau}_{j\,\min}^k$ is the joints lower limitation, $\boldsymbol{\tau}_{j\,\max}^k$ is the joints upper limitation.

### A4.4 Cost

At the end of programming, its function of the whole body trajectory is to realize the motion learned from RL framework, we implement the cost as :

$$L_i(\mathbf{x}_i, \mathbf{u}_i) = \|\mathbf{q}_i^u - \mathbf{q}_i^*\|^2 + \|\mathbf{u}\|^2 \tag{A18}$$

the term $\|\mathbf{q}_i^u - \mathbf{q}_i^*\|^2$ is for merging the gap between RL trajectory and actually feasible trajectroy, $\mathbf{q}_i^u$ is the upper body trajectory from RL, $\mathbf{q}_i^*$ is the upper body trajectory from whole body optimization, $\|\mathbf{u}\|^2$ for reduce the energy of the whole motion.

## A5 Motion Library

We constructed a motion library to house the learned whole-body skills as well as the action and condition nodes used to construct the task graph. The motion library includes information about the skills fed to the LLM, as well as the control code corresponding to each skill. The following Fig. A2, A3 shows the action skills and nodes inside the motion library that LLM can choose to invoke to construct the task graph.

```
### Action Node ###

<HomingPose>: 'name'='homing_pose'; 'type'=general; 'label'=start the robot to a initial position;
'description'=control the robot to power up and back to the initial robot pose.

<FindObject>: 'name'='find_object'; 'type'=general; 'label'=look around for object;
'description'=control the robot to turn on the head camera, and rotates itself to find 'object' and
acquire its 3D position.

<MoveTarget>: 'name'='move_target'; 'type'=wheel; 'label'=approach to target with wheels;
'description'=control the robot to approach to the target location using wheel motion (require knowing
3D position of 'target').

<WalkTarget>: 'name'='walk_target' ;'type'=leg; 'label'=approach to target with legs;
'description'=control the robot to approach to the target location using leg motion (require knowing
3D position of 'target').

<ObjectDetect>: 'name'='object_detect'; 'type'=general; 'label'=object detection and pose estimation;
'description'=using the head camera to detect and estimate the position and pose of the
'target_object'.

<ObjectPlace>: 'name'='object_place'; 'type'=general; 'label'=place object to a target position;
'description'=control the robot to put the object to a target position. (require knowing 'target' 3D
position).

<OpenDoor>: 'name'='door_open'; 'type'=general; 'label'=open the door; 'description'=control the robot
to open the door. (require knowing 3D position of 'door').

<SinglePick>: 'name'='single_arm_pick'; 'type'=single_arm; 'label'=grasp object and pick it up;
'description'=control the robot to grasp the target object with right arm, and pick it up (require
knowing 'target_object' position and pose).

<DualPick>: 'name'='dual_arm_pick'; 'type'=dual_arm; 'label'=hold object with dual arms and pick it
up. 'description'=control the robot to hold the target object with dual arms, and pick it up (require
knowing 'target_object' position and pose).

<OpenDrawer>: 'name'='open_drawer'; 'type'=general; 'label'=open the drawer. 'description'=control the
robot to open the drawer. (require knowing 3D position of 'drawer').
```

Figure A3: Action nodes in the motion library, where the blue nodes are based on learned whole-body motion skills.

```
### Condition Node ###

<Distance>: 'name'='object_in_reach'; 'type'=general; 'label'=is object in reach;
'description'=measure the distance from the object to the robot. if it is larger than 80cm then return
to fail. (require knowing 'object' 3D position)

<WhetherSingleArm>: 'name'='whether_single_arm'; 'type'=general; 'label'=select robotic morphology
based on manipulation task; 'description'=apply VLM to reason whether to use a single arm or dual arm
to manipulate object, can be used to make decisions before picking actions.

<WhetherWheelMove>: 'name'='whether_wheel_move'; 'type'=general; 'label'=select robotic morphology
based on locomotion task; 'description'=apply VLM to reason whether to use wheel or leg to move, can
be used to make decisions before locomotion actions.

<IsActionSuccess>: 'name'='is_action_completed': 'type'=general; 'label'=reason about the success of
the action; 'description'=apply VLM to reason whether the previous manipulation action is successful,
if not, repeat the action one time in behavior tree.
```

Figure A4: Condition nodes with different functions in the motion library.

## A6 Motion Morphology Selection

In this section, we show the task scenarios used for the motion morphology selection experiments.

### A6.1 Manipulation Scenarios

For the robot manipulation morphology selection experiments included six simulated and four real-world scenarios. We conducted ten morphology selections for each scenario, and before each trial, the positions and poses of the objects in the scenarios were reset. We applied the same prompts for all manipulation morphology selections, with the instructions for each scenario shown in Fig. A5.

### A6.2 Locomotion Scenarios

The robot locomotion morphology selection experiments included six simulated and four real-world scenarios, as shown in Fig.A6. We conducted ten morphology selections for each scenario, and before each trial, the positions of the robot and obstacles in the scenarios were reset. We applied the same prompts for all locomotion morphology selections.

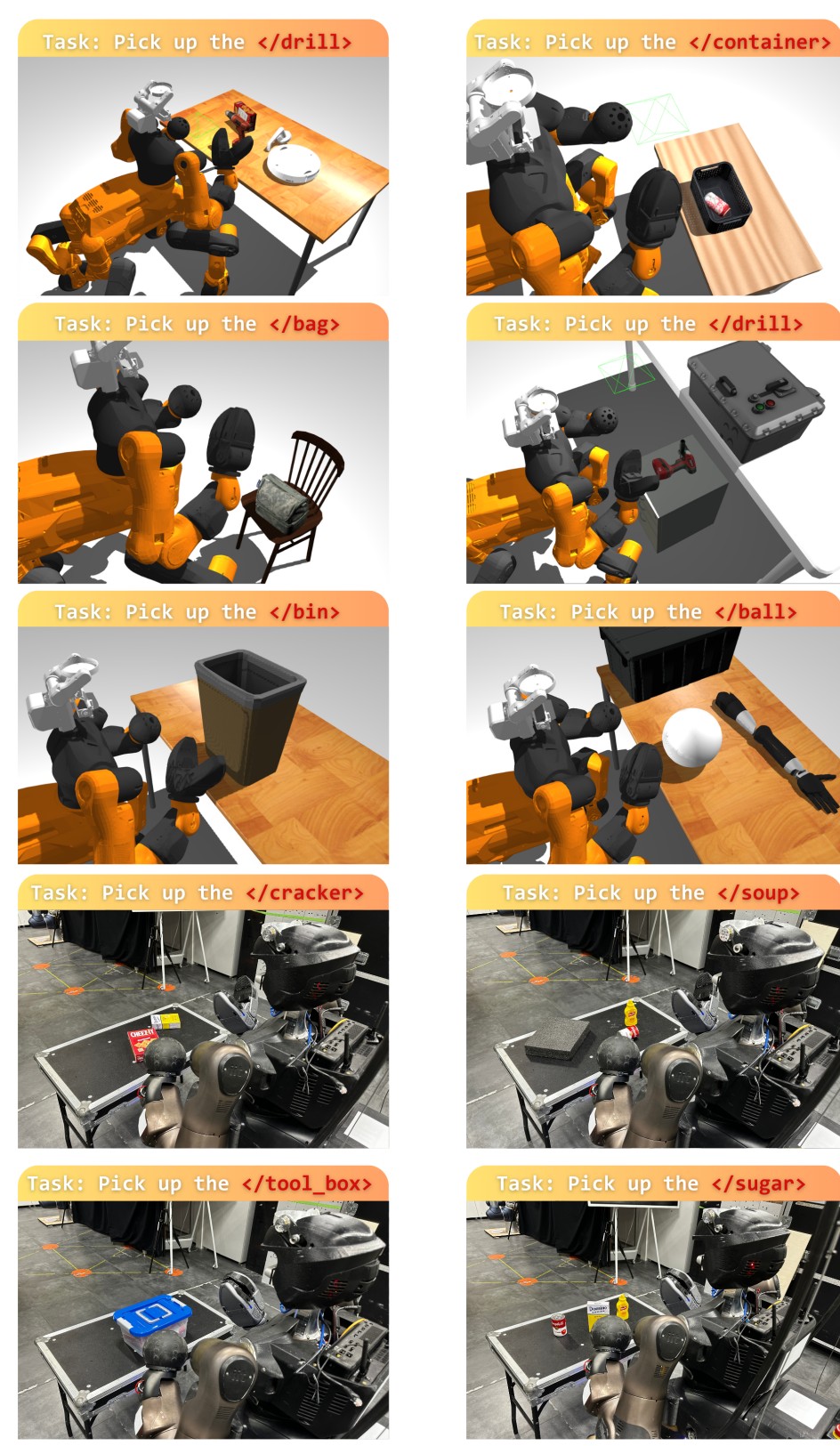

Figure A5: Task scenarios for manipulation morphology selection experiments.

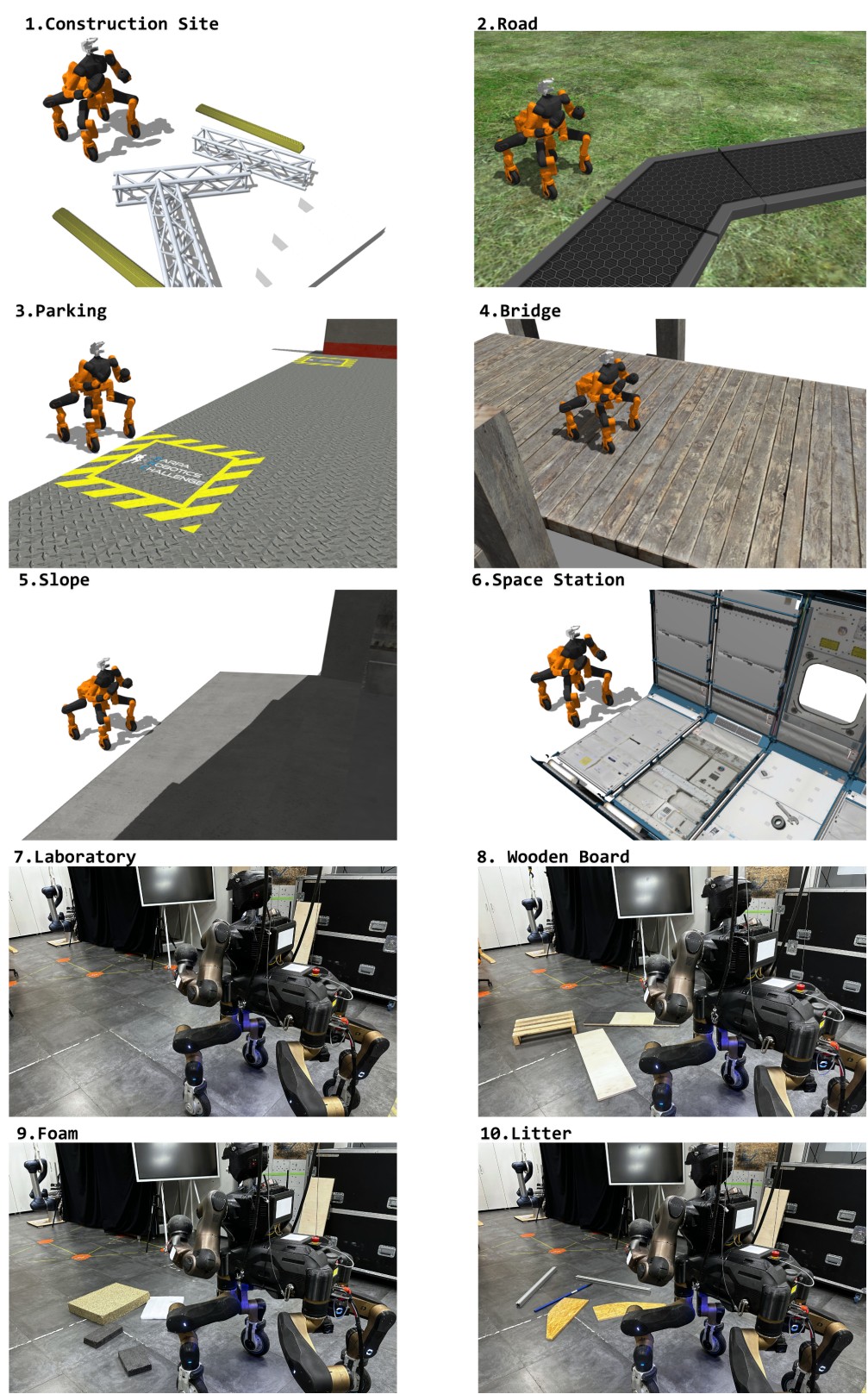

Figure A6: Task scenarios for locomotion morphology selection experiments.

### A6.3  VLM Prompts

The prompt words used for the motion morphology selector are shown in the figures, where the prompt words for manipulation morphology selector will be fed into the VLM along with the received textual task instructions from the Behavior Tree.

The motion morphology selector are packaged as one of the functions in 'User Input' module and it turned 'off' by default. When it needs to be invoked in task planning, it must be enable in 'Function Options' or specified to be set to 'on' when inputting the task instructions.

```
"Suppose you are a humanoid robot and you have two arms, the right arm has a claw gripper
as the end-effector."
"You have two ways to manipulate object: single arm manipulation and dual arm
manipulation."
"You have a camera on your head that can see the object and the environment."
"And you can choose the manipulation method based on the object and task instruction
reasoning."
"When you are doing single arm manipulation, your claw jaw gripper can open up to 10cm."
"You only have the ability to use the jaws to pick up object, regardless of other skillful
grasping methods"
"meaning that you cannot use single arm grasping if the size of the object exceeds the size
of the jaw'opening and closing,"
"or if the object current pose is not capable for grasping with one hand."
"For example: if the given task is 'pick up the drill', and the image shows that a drill is
on the desk."
"Then you will choose to use single arm manipulation, because the size of the drill in the
image can be manipulated with single arm"
"Now you receive the image from the camera and the task below, please answer whether to use
'single arm' or 'dual arm' to do the manipulation."
"(Please answer with only 'single' or 'dual')"
```

Figure A7: Prompts used for Manipulation Morphology Selection.

```
"Suppose you are a robot and you have two ways to move: legs and wheels. "
"You have a depth camera that can obtained the 2D image and point cloud of
the road in front you."

"Now you have to pass the road in front you and here is the 2D image of the
road, and the down sampled point cloud "

"Please choose whether the road should be passed with legs or wheels."
"(Note that wheels are used when the road ahead is flat, or a slope, or the
maximum height of the obstacles is lower than 5 cm."
"And legs should be used when there are obstacles or wooden planks 'maximum
height higher than 5 cm'. )"
"Determine which type of movement the robot should use to pass through the
roadway. "
"(Please answer with only 'leg' or 'wheel', and the data below is the point
cloud)"
```

Figure A8: Prompts used for Locomotion Morphology Selection.

## A7 User Input

The 'User Input' is the module that links the instructor to the language model and contains predefined prompts for initializing the language system environment and limiting the model output, as well as an interface for accepting task commands sent from the user side.

### A7.1 Basic Prompts

Basic prompts provide a description of the task context and robot characteristics, as well as an explanation of user commands and output formatting requirements. As shown below:

```
###Basic Prompts###
"You are now a robot controller, please output a XML file for
constructing a behavior tree to control the robot under the
requirements and given task."
"The robot you control is a centaur like robot, with a humanoid
upper body and four legs, each leg has a wheel at the bottom."
"The robot has two arms, with a claw gripper on the right arm.
It can manipulate objects with two ways: single-arm manipulation
and dual-arm manipulation."
"The robot has two modes of movement: wheel motion and leg motion.
The robot default manipulation and locomotion modes are
'single arm' and 'wheel'."
"The robot has two depth cameras: one located on the head to view
objects, and one on the waist to view the road and terrain ahead."
```

### A7.2 Function Options

We designed a number of functions for the robot and packaged them into condition nodes for selective invocation by the LLM during the planning of the task. These functions include: 'Manipulation Morphology Selector', 'Locomotion Morphology Selector', 'Failure Detection and Recovery'. We add the descriptions of these functions acting as 'Function Options' inside the 'User Input', and set all functions to 'off' state by default. When the instructor expects a function to be added during a task planning, it can be manually set to 'on' or include a declaration to use the function in the instruction.

```
###Function Options###
"The robot has the following functions, all of which are 'off' by
    default."
"When a function is 'on', it need to be involved in planning for the
    given task, and when it is 'off', it should not be used."
"Functions: "

"1. 'manipulation_mode_selector': this function allows the robot to
    add the condition node <WhetherSingleArm> to the planning of
    BehaviorTree, which is used to determine whether the current
    manipulation task should use the 'single_arm' or 'dual_arm' type
    of action."

"2. 'locomotion_mode_selector': this function allows the robot to add
    the condition node <WhetherWheelMove> to the planning of the
    behavior tree, which is used to determine whether the current
    locomotion task should use the 'wheel' or 'leg' type of action."

"3. 'detection_recovery': this allows the robot to add the condition
    node <IsActionSuccess>, which is used to determine whether the
    previous action has been successfully completed and, if not, to
    employ a recovery mechanism that repeat the action."
```

### A7.3 User Interface

The user interface is responsible for accepting task commands from the instructor and combining them with pre-defined prompt for input to the LLM. The complete user input is as follows.

`User Interface:` hy-motion.github.io/prompt/user_input.ini

`Motion Library:` hy-motion.github.io/prompt/motion_library.ini

`Basic Prompts:` hy-motion.github.io/prompt/basic_prompt.ini

`Function Options:` hy-motion.github.io/prompt/Function_options.ini

## A8 Task Planning with LLM

After receiving the prompts from 'User Input', the LLM output a hierarchical task graph that contains a series of nodes and actions for accomplishing the task. The task graph is saved in an .xml file and serves as a framework for constructing the Behavior Tree that guides the robot's actions. Below we show the detail of experiments in 'Tasks with human instructions' part of Sec. 4.3. For each task, we present the task graph generated by LLM, and the Behavior Tree constructed from it.

```
Input: Open the drawer and pick up the drill.
```

```xml
<root BTCPP_format="4" >
 <BehaviorTree ID="MainTree">
 <Sequence name="root">
 <FindObject name="find_object" goal="{DrawerPosition}"/>
 <Fallback>
 <Distance name="object_in_reach" message="{DrawerPosition}"/>
 <MoveTarget name="move_target" message="{DrawerPosition}"/>
 </Fallback>
 <OpenDrawer name="open_drawer"/>
 <ObjectDetect name="object_detect" goal="{DrillPose}"/>
 <SinglePick name="single_arm_pick" message="{DrillPose}"/>
 </Sequence>
 </BehaviorTree>
</root>
```

Figure A9: Task planning of 'Open drawer and pick object'.

```
Input: Find the door and open it.
```

```xml
<root BTCPP_format="4">
 <BehaviorTree ID="MainTree">
 <Sequence name="root">
 <FindObject name="find_object" goal="{DoorPosition}"/>
 <Fallback>
 <Distance name="object_in_reach" message="{DoorPosition}"/>
 <MoveTarget name="move_target" message="{DoorPosition}"/>
 </Fallback>
 <OpenDoor name="door_open"/>
 </Sequence>
 </BehaviorTree>
</root>
```

Figure A10: Task planning of 'Approach and open door'.

```
Input: Pick up the cracker and put it into the box.
```

```
<root BTCPP_format="4">
 <BehaviorTree ID="MainTree">
 <Sequence name="root">
 <FindObject name="find_object" goal="{CrackerPosition}"/>
 <Fallback>
 <Distance name="object_in_reach" message="{CrackerPosition}"/>
 <MoveTarget name="move_target" message="{CrackerPosition}"/>
 </Fallback>
 <SinglePick name="single_arm_pick" message="{CrackerPosition}"/>
 <FindObject name="find_object" goal="{BoxPosition}"/>
 <Fallback>
 <Distance name="object_in_reach" message="{BoxPosition}"/>
 <MoveTarget name="move_target" message="{BoxPosition}"/>
 </Fallback>
 <ObjectPlace name="object_place" message="{BoxPosition}"/>
 </Sequence>
 </BehaviorTree>
</root>
```

Figure A11: Task planning of 'Pick and place'.

```
Input: Pick up the box and put it on the table.
       ('manipulation_mode_selector'=on)
```

```
<root BTCPP_format="4">
 <BehaviorTree ID="MainTree">
 <Sequence name="root">
 <FindObject name="find_object" goal="{BoxPosition}"/>
 <Fallback>
 <Distance name="object_in_reach" message="{BoxPosition}"/>
 <MoveTarget name="move_target" message="{BoxPosition}"/>
 </Fallback>
 <ObjectDetect name="object_detect" goal="{BoxPose}"/>
 <Fallback>
 <Sequence>
 <WhetherSingleArm name="whether_single_arm" message="{BoxPose}"/>
 <SinglePick name="single_arm_pick" message="{BoxPose}"/>
 </Sequence>
 <DualPick name="dual_arm_pick" message="{BoxPose}"/>
 </Fallback>
 <FindObject name="find_object" goal="{TablePosition}"/>
 <Fallback>
 <Distance name="object_in_reach" message="{TablePosition}"/>
 <MoveTarget name="move_target" message="{TableGeneratorPosition}"/>
 </Fallback>
 <ObjectPlace name="object_place" message="{TablePosition}"/>
 </Sequence>
 </BehaviorTree>
</root>
```

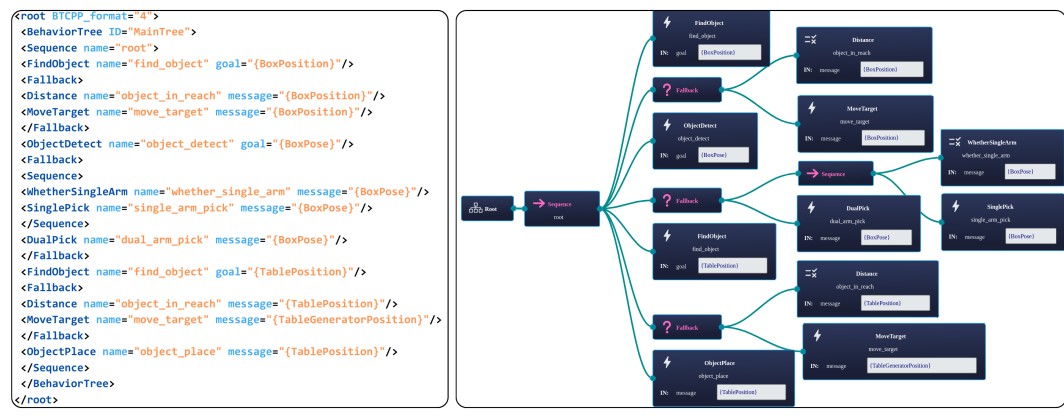

Figure A12: Task planning of 'Dual-arm pick place'.

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
