# OpenReview forum: "HYPERmotion: Learning Hybrid Behavior Planning for Autonomous Loco-manipulation"
_robot-learning.org/CoRL/2024/Conference — CoRL 2024_

### Official Review · Reviewer_3ACE · 2024-07-20
**HYPERmotion: Learning Hybrid Behavior Planning for Autonomous Loco-manipulation**

**Originality:** 2
**Technical Quality:** 3
**Clarity Of Presentation:** 4
**Potential Impact:** 2
**Recommendation:** 3
**Confidence:** 5

**Review:**

Paper Strengths:
1. A complex integrated system tying multi components like RL, LLM and VLMs.
2. Demonstration of LLMs capability to compose of learned skills.
3. Evaluation in both simulation and the real robot system.

Weaknesses:
1. Limited quantitative evaluation and comparison to baseline methods.
2. Lack of detail on some technical aspects, e.g. the whole-body motion optimization.
3. Disjoint upper body and lower body control and non-general approach for navigation using AprilTags.
4. A large number of system components making it difficult to test, reproduce and deploy.
5. Limited discussion of failure cases.


Suggestions:
- (Lines 19-20) "Experiments in simulation and real-world show that learned motions can efficiently adapt to new tasks, demonstrating high autonomy from free-text commands in unstructured scenes." - while experiments are described, the paper doesn't provide quantitative metrics for adaptation efficiency / capabilities, making this strong claim difficult to fully verify.
- (line 272-274) Please support claim like "We found that language-based behavior planner exhibits greater versatility and adaptability to more complex tasks compared to existing methods." with experimental data - currently the paper has no comparison with other methods
- (line 83 - typo) - teleportation -> teleoperation

Overall, this work presents an interesting approach for leveraging large model in a zero shot fashion for enabling more general and flexible behavior in humanoid robots. The integration of learning-based motion generation with language model planning is particularly promising. However, the experimental evaluation is somewhat limited in scope and more extensive comparisons and analysis would strengthen the paper. Nonetheless, this represents a meaningful step towards more capable and general-purpose humanoid robots.

**Quality Of The Limitations Section:**

3

**Questions For Rebuttal:**

1. Can you provide more quantitative comparisons to baseline methods, e.g. planning without language models using a custom decision tree? It is hard to grasp the complexity of the domain for the limited text description of the tasks and how much role do LLMs and VLMs actually play.

2. Can you elaborate on failure cases or limitations observed in your experiments? Are there particular types of tasks or scenarios where the system struggles?

3. Can you provide more details on the whole-body motion optimization process? How do you ensure dynamic feasibility?

4. How did you approach the question of what primitives to include in the learning library and how easy it is to acquire a specific primitive?

5. Can you give a comment on what component of the system could be: combined, remove, added? and why?

**Robotics Focus:**

4

**Summary Of Paper:**

This paper presents a framework for enabling humanoid robots to learn, select, and plan behaviors for loco-manipulation tasks. The approach combines RL for generating a motion primitive library, LLM for high-level task planning, and VLM for selecting appropriate robot morphologies. The authors demonstrate the system on a centaur-like humanoid robot in both simulated and real-world environments.

**Summary Of Recommendation:**

The proposed approach shows promise in zero-shot generalization to new tasks. While it represents a step forward in autonomous loco-manipulation, it lacks comprehensive quantitative comparisons and detailed technical explanations. The system's complexity raises concerns about usability and reproducibility. Despite these weaknesses, the integration of language models with robotic control is timely and potentially impactful. I recommend a weak accept, contingent on addressing these issues, particularly by providing more rigorous evaluation and critical analysis of limitations.

---

### Official Review · Reviewer_7D92 · 2024-07-21
**This paper showcases a robot system for long-horizon tasks from human commands, highlighting technical integration but lacking detailed experiments.**

**Originality:** 3
**Technical Quality:** 4
**Clarity Of Presentation:** 3
**Potential Impact:** 3
**Recommendation:** 3
**Confidence:** 4

**Review:**

In this paper the authors present an fully integrated system demonstration that autonomous execution of long horizon tasks from human language commands on a platform capable of locomotion and manipulation.

The robot demonstration on hardware are compelling. They relies on a great effort of integration of novel techniques (VLM, LLM, etc.) and classical ones whole-body controller, behavior trees, etc. It is a great achievement to integrate all of these technical building blocks together in a single autonomous system.

The authors offer valuable insight into the performance of their system with a breakdown of the failure modes between different modules. They also provide extensive details about their modular pipeline. I would say that this paper reads more like a technical report as it describes important technical choices. However, except for Fig 6, the paper does not focus on justifying these choices through thorough experiments, baseline comparisons, and ablations studies.

Yet I still think this work is an informative data point for other researchers to understand how far we can go with the current tools at our disposal.

Finally, the authors could further improve the clarity of the paper by concisely stating and pinpointing the contribution of the manuscript in the introduction section.

**Quality Of The Limitations Section:**

3

**Questions For Rebuttal:**

The authors could further improve the clarity of the paper by concisely stating and pinpointing the contribution of the manuscript in the introduction section.

**Robotics Focus:**

4

**Summary Of Paper:**

In this paper the authors present an fully integrated system demonstration that autonomous execution of long horizon tasks from human language commands on a platform capable of locomotion and manipulation.

**Summary Of Recommendation:**

I would recommend a weak accept since this work is valuable for the research community but its contribution lies in system integration rather than the development of novel techniques.

---

### Official Review · Reviewer_Ec5Y · 2024-07-23

**Originality:** 2
**Technical Quality:** 3
**Clarity Of Presentation:** 4
**Potential Impact:** 2
**Recommendation:** 3
**Confidence:** 4

**Review:**

Strengths:

- The paper is in general clear to understand and figures are well-made to assist readers in understanding the framework.
- The real-world demonstration is impressive and refreshing to see.

Weaknesses:

- It is unclear whether the paper provides new insights for the community and it lacks a central focus. Specifically, the paper seems to leverage LLMs / VLMs in many components in the pipeline, most of which have been heavily studied in prior works, while using standard RL to train common skills. As a result, none of these may qualify for a unique contribution of the paper. Given the extent of the framework, many important details are missing or simply do not have the space to be discussed in the main paper.
- Similarly, given the extent of the framework, the evaluations appear to be severely lacking and missing important details. For example, what are the success rates of the learned RL policies alone? What is the evaluation procedure (e.g., objects and their initial and goal randomization)? What is the scope of the long-horizon tasks that are using LLMs as planners?

**Quality Of The Limitations Section:**

3

**Questions For Rebuttal:**

See weakness section above.

**Robotics Focus:**

4

**Summary Of Paper:**

The paper tackles the loco-manipulation problem for a dual-arm quadruped robot. Specifically, the paper focuses on an LLM-based pipeline with various components to complete loco-manipulation tasks such as articulated object manipulation, grasping, and traversing different terrains, where low-level skills are trained with RL and provided in a skill library. Demonstrations are shown on real-robot hardware.

**Summary Of Recommendation:**

While the paper provides impressive real-world demonstrations, it appears that it lacks a central focus to provide insightful contributions to the community. Additionally, given the large scope of the claimed framework, the evaluations and their details appear to be lacking as well. Therefore, I’m withholding a higher recommendation.

---

### Author Rebuttal · Authors · 2024-08-08

Dear Area Chair and Reviewers,

In the CoRL_rebuttal.zip file, we have added the updated manuscript (CoRL_rebuttal.pdf), the updated appendix (Appendix_rebuttal.pdf), a description of experiments, and the supplement video.

Thanks!

---

### Decision · Program_Chairs · 2024-09-04

**Decision:**

Accept

**Comment:**

The paper has addressed my concerns. I encourage the authors to follow Ec5Y's advice and frame it as a systems paper. Overall, I found the paper interesting, and showcases an interesting application of VLM and LLM for humanoid control in the real world.


==============================

Strengths:
- Impressive system that showcases what integration of VLM, RL, and VLM can achieve
- Real-world experiment is appreciated.

Weaknesses:
- It lacks a scientific contribution and the paper reads as a tech report that tells the reader how the system can be integrated.

The authors are encouraged to pinpoint the technical challenges they faced and how they solved those challenges.